# Pregnancy Outcomes in Females with Stage 1 Hypertension and Elevated Blood Pressure Undergoing In Vitro Fertilization and Embryo Transfer

**DOI:** 10.3390/jcm12010121

**Published:** 2022-12-23

**Authors:** Shaomin Chen, Yang Wang, Yongqing Wang, Yuan Wei, Yanguang Li, Zhaoping Li, Rong Li

**Affiliations:** 1Key Laboratory of Cardiovascular Molecular Biology and Regulatory Peptides, Key Laboratory of Molecular Cardiovascular Science, Ministry of Education, Beijing Key Laboratory of Cardiovascular Receptors Research, Department of Cardiology and Institute of Vascular Medicine, Peking University Third Hospital, Ministry of Health, Beijing 100191, China; 2Key Laboratory of Assisted Reproduction (Peking University), Beijing Key Laboratory of Reproductive Endocrinology and Assisted Reproductive Technology, Center for Reproductive Medicine, National Clinical Research Center for Obstetrics and Gynecology, Department of Obstetrics and Gynecology, Peking University Third Hospital, Ministry of Education, Beijing 100191, China; 3Department of Obstetrics and Gynecology, Peking University Third Hospital, Beijing 100191, China

**Keywords:** in vitro fertilization and embryo transfer, hypertension, hypertensive disorders in pregnancy, preeclampsia

## Abstract

**Objective:** To determine whether stage 1 hypertension and elevated blood pressure (BP), as defined by the 2017 American College of Cardiology/American Heart Association (ACC/AHA) guidelines, prior to pregnancy contributes to adverse pregnancy outcomes in females who conceived by in vitro fertilization and embryo transfer (IVF–ET). **Methods:** This retrospective cohort study involved 2239 females who conceived by IVF–ET and delivered live neonates. BPs recorded before IVF–ET were collected. Elevated BP was defined as at least two systolic BPs of 120 to 129 mmHg. Stage 1 hypertension was defined as at least two systolic BPs of 130 to 139 mmHg or diastolic BPs of 80 to 89 mmHg. **Results:** Among the females included in this study, 18.5% (415/2239) had elevated BP and 10.0% (223/2239) had stage 1 hypertension. Multiple logistic regression analysis showed that females with stage 1 hypertension had higher risks of hypertensive disorders in pregnancy (HDP) [adjusted odds ratio (aOR) 1.65; 95% confidence interval (CI) 1.16–2.35] and preeclampsia (aOR 1.52; 95% CI 1.02–2.26) than normotensive females. However, the risks of HDP (aOR 0.88; 95% CI 0.64–1.21) and preeclampsia (aOR 0.83; 95% CI, 0.57–1.20) in females with elevated BP were not significantly different from those in normotensive females. The females were then categorized into five groups by systolic and diastolic BP; females with systolic BP of 130 to 139 mmHg or diastolic BP of 85 to 89 mmHg had significantly increased risks of HDP and preeclampsia. **Conclusion:** Stage 1 hypertension before IVF–ET was an independent risk factor for HDP and preeclampsia.

## 1. Introduction

In vitro fertilization and embryo transfer (IVF–ET) is performed worldwide [1,2]. In mainland China, the number of infants born through IVF–ET accounted for 1.69% of the total number of live infants in 2016 [3]. It has been established that IVF–ET is associated with adverse pregnancy outcomes, including preeclampsia, a neonate being small for gestational age, stillbirth, placental abruption, cesarean delivery, prematurity and neonatal death [4,5,6,7]. Females pregnant by IVF–ET are older and more likely to have chronic hypertension than females who have spontaneously conceived [4,5].

Chronic hypertension prior to pregnancy is also a significant risk factor for adverse pregnancy outcomes, [8] and IVF–ET may further augment these risks in hypertensive females [4]. One study showed that the risk of preeclampsia in hypertensive females with IVF–ET pregnancies was 10% higher than that in hypertensive females with spontaneous pregnancies [4]. Chronic hypertension is defined as blood pressure (BP) ≥ 140/90 mmHg, pre-existing before pregnancy, or occurring before 20 weeks of pregnancy, according to obstetrical guidelines [8,9]. However, the 2017 American College of Cardiology/American Heart Association (ACC/AHA) guidelines recommended a lower threshold (130/80 mmHg) for hypertension in non-pregnant adults [10]. Stage 1 hypertension was defined as a systolic BP of 130 to 139 mmHg and/or a diastolic BP of 80 to 89 mmHg, while elevated BP was defined as a systolic BP of 120 to 129 mmHg and a diastolic BP of <80 mmHg. However, these definitions have not been adopted in pregnant females.

During the past five years, several retrospective cohort studies and a meta-analysis have shown that females with stage 1 hypertension and elevated BP before or during early pregnancy are at a higher risk of developing hypertensive disorders of pregnancy (HDP) and preeclampsia than females with normal BP [11,12,13,14,15,16,17,18,19,20,21] Whether females with stage 1 hypertension and elevated BP before IVF–ET are more likely to develop pregnancy complications than females with normal BP is unknown. Thus, the purpose of this study was to evaluate the risk of adverse pregnancy outcomes in females with stage 1 hypertension and elevated BP who conceived by IVF–ET.

## 2. Method

### 2.1. Study Population

This retrospective cohort study was conducted at the Reproductive Medical Center of the Peking University Third Hospital, a tertiary university hospital and a Center of Excellence in Reproductive Medicine in China. We included females who conceived by IVF–ET and delivered live neonates between 1 January 2017 and 31 December 2020 in our hospital. The exclusion criteria were systolic BP of ≥140 mmHg, diastolic BP of ≥90 mmHg, or taking antihypertensive medications before IVF–ET. For females with more than one birth during the study period, only data from the first pregnancy were analyzed. 

The following data were collected from patients’ electronic medical records: pre-pregnancy body mass index (BMI); histories of diabetes mellitus and autoimmune diseases; time of delivery; basal concentrations of follicle-stimulating hormone (FSH), luteinizing Hormone (LH), estradiol (E2) and androstenedione (A); etiology of infertility; protocol of IVF; type of embryo transfer; stage of transferred embryo; number of embryos transferred.

### 2.2. Blood Pressure Measurement and Definition

Using the patients’ electronic medical records, we examined all BP measurements before IVF–ET. At least three BPs were routinely measured by trained heath care workers using an Omron automated sphygmomanometer. The females were categorized as having normal BP, elevated BP, or stage 1 hypertension, according to the 2017 ACC/AHA guidelines [10]. Elevated BP was defined as at least two systolic BPs of 120 to 129 mmHg. Stage 1 hypertension was defined as at least two systolic BPs of 130 to 139 mmHg or diastolic BPs of 80 to 89 mmHg.

### 2.3. Outcomes

The maternal outcomes were HDP and preeclampsia. HDP was a composite of gestational hypertension, chronic hypertension, preeclampsia, eclampsia, and hemolysis, elevated liver enzymes, and low platelet count (HELLP) syndrome. In our institution, the definitions of HDP follow the American College of Obstetricians and Gynecologists guidelines [8,22]. Gestational hypertension was defined as a systolic BP of ≥140 mmHg or a diastolic BP of ≥90 mmHg, on two occasions at least 4 h apart after 20 weeks of gestation without proteinuria or severe features. Since females with pre-pregnancy systolic BP of ≥140 mmHg, or diastolic BP of ≥90 mmHg, were excluded, chronic hypertension was defined as BP of ≥140/90 mmHg before 20 weeks of pregnancy, with or without preeclampsia, in this study. Preeclampsia was defined as a systolic BP of ≥140 mmHg or a diastolic BP of ≥90 mmHg on two occasions at least 4 h apart after 20 weeks’ gestation, with proteinuria or with severe features (thrombocytopenia, renal insufficiency, impaired liver function, pulmonary edema, new-onset headache unresponsive to medication, or visual symptoms).

The neonatal outcomes were preterm birth, low birth weight and the neonate being small for gestational age. Preterm birth was defined as birth at <37 completed weeks of gestation. Low birth weight was defined as a birth weight <2500 g. A neonate being considered small for gestational age was defined as a birth weight below the 10th centile for the gestational age and sex of the baby.

### 2.4. Statistical Analyses

The Shapiro–Wilk test was performed to assess the normality of the data distribution. Continuous data with a normal distribution were expressed as mean ± standard deviation, and the comparisons among the three groups were conducted by the ANOVA test. Categorical data were expressed as number (percentage) and the comparisons among the three groups were conducted by the Chi-square test. To analyze whether stage 1 hypertension and elevated BP before pregnancy were associated with higher risks of adverse outcomes, we performed a multiple logistic regression analysis and adjusted for potential confounders selected a priori, including maternal age, pre-pregnancy BMI, parity, twin pregnancy, history of diabetes mellitus, etiology of infertility, protocol of IVF, type of embryo transfer, stage of transferred embryo, and number of embryos transferred [15]. The females were then categorized into five groups by their systolic and diastolic BP, and the relationship between BP and the risk of adverse outcomes was analyzed using multiple logistic regression analysis. The analyses were performed using SPSS version 23.0 (IBM Corp., Armonk, NY, USA), and a *p*-Value of < 0.05 was considered statistically significant.

## 3. Results

### 3.1. The Characteristics of Females in the Normal BP, Elevated BP and Stage 1 Hypertension Groups

In total, 79,576 females underwent IVF–ET between 1 January 2017 and 31 December 2020 in our hospital. Among them, 2285 females delivered live neonates in our hospital and were included in this study and 24,136 females delivered in other clinics and hospitals and were not included. Females with systolic BP of ≥140 mmHg, diastolic BP of ≥90 mmHg, or taking antihypertensive medications before IVF–ET were excluded (*n* = 46). Finally, a total of 2239 females were included for analysis (Figure 1).

In total, 18.5% (415/2239) of women in the study population had elevated BP and 10.0% (223/2239) had stage 1 hypertension. Twin pregnancy was present in 29.8% (668/2239) of females, and singleton pregnancy in 70.2% (1571/2239). Table 1 summarizes the clinical characteristics of the study population, according to pre-IVF BP levels. The pre-pregnancy BMI was significantly different among the normal BP, elevated BP and stage 1 hypertension groups (21.9 ± 2.9, 22.3 ± 3.2 and 22.8 ± 3.1 kg/m^2^, respectively; *p* < 0.001). Females with elevated BP and stage 1 hypertension had a significantly higher pre-pregnancy BMI than females who were normotensive (*p* < 0.05). There were no significant differences in maternal age, parity, twin pregnancy, histories of diabetes and autoimmune diseases, time of delivery, basal concentrations of FSH, LH, E2 and A, etiology of infertility, protocol of IVF, type of embryo transfer, stage of transferred embryo or number of embryos transferred among the three groups (Table 1).

### 3.2. Adverse Pregnancy Outcomes in the Normal BP, Elevated BP and Stage 1 Hypertension Groups

Overall, 14.9% (333/2239) of females developed HDP, and 11.0% (247/2239) of females had preeclampsia. The incidences of HDP and preeclampsia were significantly different among the three groups (HDP: 14.1%, 14.0% and 22.4%, respectively, *p* = 0.004; and preeclampsia: 10.7%, 9.6% and 16.1%, respectively, *p* = 0.031). Subgroup analysis revealed that patients with stage 1 hypertension were significantly more likely to develop HDP and preeclampsia than patients in the elevated BP group and the normotension group (*p* < 0.05 for both HDP and preeclampsia). However, the elevated BP group did not have a significantly higher risk of developing HDP or preeclampsia than the normal BP group. No apparent differences were found in preterm birth, low birth weight or neonates being small for gestational age among the three groups (Table 2).

Multiple logistic regression analysis showed higher likelihoods of developing HDP [adjusted odds ratio (aOR) 1.65; 95% confidence interval (CI) 1.16–2.35] and preeclampsia (aOR 1.52; 95% CI 1.02–2.26) in females with stage 1 hypertension than in normotensive females. However, the risks of HDP (aOR 0.88; 95% CI 0.64–1.21) and preeclampsia (aOR 0.83; 95% CI, 0.57–1.20) in females with elevated BP were not significantly different from those in normotensive females (Table 3). There were no significant differences in the risk of preterm birth, low birth weight or neonates being small for gestational age among the three groups (Table 3).

### 3.3. Relationships between BP and Maternal Outcomes

Maternal outcomes were assessed according to the BP categories. Females with systolic BP of 130 to 139 mmHg, or diastolic BP of 85 to 89 mmHg, had a significantly increased risk of HDP and preeclampsia (Table 4, Figure 2). For females with systolic BP of <130 mmHg, or diastolic BP of <85 mmHg, the increase in systolic BP, or diastolic BP, was not associated with a significant increase in the risk of adverse outcomes (Table 5, Figure 2).

## 4. Discussion

This cohort study included 2239 females who had become pregnant by IVF–ET and delivered live neonates. The relationship between BP before IVF–ET and adverse pregnancy outcomes was investigated. The main findings of this study were as follows: (1) stage 1 hypertension, as defined by the 2017 ACC/AHA guidelines, was independently associated with an increased risk of HDP and preeclampsia; (2) there was no significant association between elevated BP, as defined by the 2017 ACC/AHA guidelines, and adverse pregnancy outcomes; (3) females with systolic BP of 130 to 139 mmHg, or diastolic BP of 85 to 89 mmHg, had a significantly increased risk of HDP and preeclampsia.

After the ACC/AHA lowered the threshold for the definition of hypertension from 140/90 mmHg to 130/80 mmHg, the prevalence of hypertension increased two-fold among young females [23]. McLaren et al. [24] analyzed births before and after establishment of the ACC/AHA guidelines and reported that the prevalence of chronic hypertension in pregnancy increased from 1.61% to 2.21%. Our data showed that up to 10.0% of females were identified as having stage 1 hypertension before IVF. The incidence of stage 1 hypertension in pregnancy was much higher than that in the study by McLaren et al. [24] This might be because females who conceived using IVF were older and had more comorbidities, including obesity, than females who spontaneously conceived.

It has been established that cardiovascular risk increases as BP moves from normal to elevated levels and stage 1 hypertension [25]. The hazard ratios for cardiovascular diseases (CVDs) ranged from 1.1 to 1.5 for the comparison of elevated BP versus normal BP, and from 1.5 to 2.0 for the comparison of stage 1 hypertension versus normal BP. The threshold for chronic hypertension remained at 140/90 mmHg in the obstetrical guidelines because the clinical significance of stage 1 hypertension in the pre-pregnancy period had not been well examined [8]. However, numerous studies and meta-analyses have implied a relationship between stage 1 hypertension and adverse pregnancy outcomes [11,12,13,14,15,16,17,18,19,20,21]. In a study by Greenberg et al., [18] the risk of HDP increased from 4.2% in females with normal BP to 10.9% in females with stage 1 hypertension (aOR, 2.54; 95% CI, 2.09–3.08). Wu et al. [16] reported similar results; females with stage 1 hypertension had more than twice the risk of developing HDP compared with normotensive females. A recently published meta-analysis showed that females with stage 1 hypertension had three times the risk of developing preeclampsia compared with females who had normal BP [20]. Our study focused on females who had become pregnant by IVF–ET, and we found that stage 1 hypertension was associated with a significantly higher risk of HDP and preeclampsia. We categorized the study subjects into five groups by systolic BP and diastolic BP separately, and found that having a systolic BP of 130 to 139 mmHg, or a diastolic BP of 85 to 89 mmHg, increased the risk of HDP and preeclampsia.

Several studies have shown that elevated BP, as defined by the ACC/AHA guidelines, is also associated with a higher risk of HDP and preeclampsia [14,15,18,19,21]. Reddy et al. [15] found that females with elevated BP were 2.45 times (95% CI, 1.74–3.44) more likely to develop preeclampsia than normotensive females. Greenberg et al. [18] reported that females with elevated BP had 1.5 times the risk of developing HDP compared with normotensive females. In a study by Jung et al., [19] lower BP was associated with a significantly lower risk of adverse pregnancy outcomes, even in regard to BP lower than 130/80 mmHg. In contrast to previous studies, our study did not show an association between elevated BP and adverse pregnancy outcomes; for females with systolic BP of <130 mmHg, or diastolic BP of <85 mmHg, and the increase in systolic BP or diastolic BP was not associated with a significant increase in the risk of adverse outcomes. There might be several reasons for the differences in these results. First, this study focused on a specific group of females who conceived by IVF–ET. Advanced age, twin pregnancy, obesity, and diabetes mellitus are usually more prevalent in this population than females who spontaneously conceive [4,5]. These females had an increased risk of developing HDP and preeclampsia, [4,5,6,7] but their elevated BPs did not further augment the risk of adverse outcomes. Second, BPs were measured before pregnancy in this study. By contrast, most of the previous studies measured BPs during early pregnancy [14,15,18,21]. BP declines during early pregnancy, [8] so females with elevated BP during early pregnancy might have had higher BP before pregnancy.

In our study, neither stage 1 hypertension, nor elevated BP, was associated with an increased risk of adverse neonatal outcomes. Although several studies have shown that stage 1 hypertension and elevated BP were independent risk factors for adverse neonatal outcomes, [13,15,16,18,19,21], a meta-analysis of 23 studies showed no significant increase in the risk of adverse neonatal outcomes in females with elevated BP and stage 1 hypertension [20].

Obesity is a risk factor for HDP and pre-eclampsia [26,27]. In a study by Bartsch et al., females with a pre-pregnancy BMI of >30 had a 2.8 times higher risk of developing preeclampsia [27]. In our study, females with elevated BP and stage 1 hypertension had a significantly higher pre-pregnancy BMI than females with normal BP. Thus, a multiple logistic regression analysis was performed to adjust for confounding factors, including BMI. This analysis showed that stage 1 hypertension was an independent risk factor for HDP and preeclampsia.

Advanced age, [28,29] twin pregnancy [30,31] and diabetes mellitus [32,33] are all independent risk factors for developing HDP and pre-eclampsia. Studies have also suggested that endometriosis, [34] male factor infertility, [35] hormone replacement therapy cycle, [36] and frozen embryo transfer [37] are associated with an increased risk of adverse pregnancy outcomes. In our study, stage 1 hypertension was independently associated with the risk of HDP and preeclampsia, after adjustment for these confounding factors.

This study had several limitations. This was a retrospective study at a single medical center. Owing to its retrospective design, we could not control how the BP readings were obtained, which may have caused inter-observer variability. Additionally, we did not collect ambulatory BP because ambulatory BP monitoring was not routinely performed before or during pregnancy. Thus, white coat hypertension, which is also a risk factor for adverse pregnancy outcomes, could not be excluded [38]. Finally, we did not collect data on cardiovascular and renal diseases that can cause adverse pregnancy outcomes because these diseases are rare in females undergoing IVF–ET.

## 5. Conclusions

Stage 1 hypertension, as defined by the 2017 ACC/AHA guidelines, before IVF–ET was an independent risk factor for HDP and preeclampsia. Prospective data are needed to determine whether more frequent follow-up of females who undergo IVF–ET, and who present with stage 1 hypertension prior to, or early in, pregnancy may be useful in identifying those at high risk for HDP and preeclampsia.

## Figures and Tables

**Figure 1 jcm-12-00121-f001:**
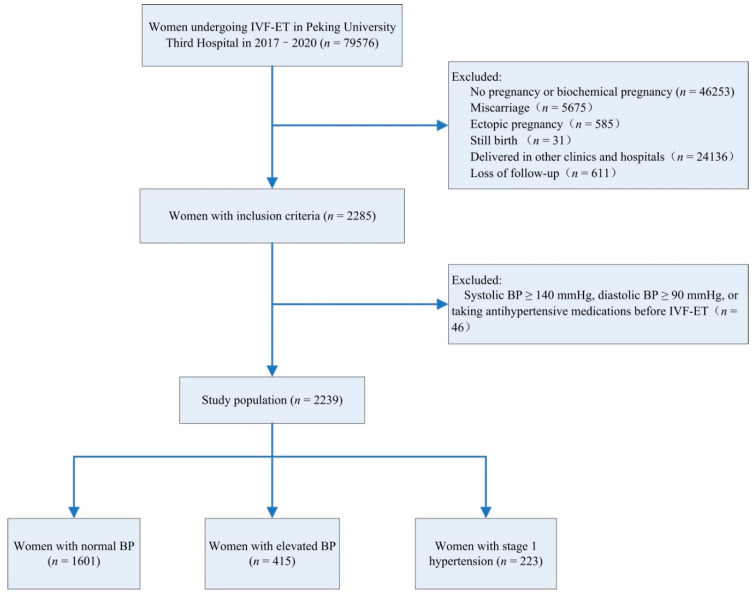
Study flowchart. BP, blood pressure; IVF–ET, in vitro fertilization and embryo transfer.

**Figure 2 jcm-12-00121-f002:**
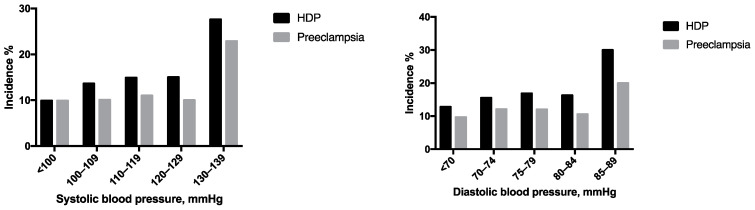
The incidence of HDP and preeclampsia at different blood pressure levels. HDP, hypertensive disorders of pregnancy.

**Table 1 jcm-12-00121-t001:** The characteristics of the females in the normal blood pressure, elevated blood pressure and stage 1 hypertension groups.

	Normal Blood Pressure (*n* = 1601)	Elevated Blood Pressure (*n* = 415)	Stage 1 Hypertension (*n* = 223)	*p*
Maternal age, years	34.3 ± 3.5	34.4 ± 3.5	34.3 ± 3.5	0.775
Pre-pregnancy body mass index, kg/m^2^	21.9 ± 2.9	22.3 ± 3.2 *	22.8 ± 3.1 *	<0.001
Primiparous, *n* (%)	1528 (95.4)	399 (96.1)	218 (97.8)	0.251
Twin pregnancy, *n* (%)	479 (29.9)	114 (27.5)	75 (33.6)	0.266
Diabetes mellitus, *n* (%)	45 (2.8)	15 (3.6)	10 (4.5)	0.331
Autoimmune diseases, *n* (%)	45 (2.8)	9 (2.2)	4 (1.8)	0.559
Time of delivery, gestational weeks	37.8 ± 2.4	37.7 ± 2.7	37.6 ± 2.5	0.484
Basal follicle-stimulating hormone, MIU/mL	6.5 ± 3.0	6.4 ± 2.6	6.4 ± 2.8	0.952
Basal Luteinizing Hormone, MIU/mL	4.3 ± 3.7	4.1 ± 3.3	4.3 ± 3.0	0.685
Basal estradiol, pmol/L	166.8 ± 91.2	176.5 ± 96.4	167.0 ± 76.2	0.182
Basal androstenedione, nmol/L	6.4 ± 3.6	6.9 ± 4.0	6.4 ± 3.5	0.182
Etiology of infertility				0.794
Tubal factor, *n* (%)	485 (30.3)	131 (31.6)	72 (32.3)	
Endometriosis, *n* (%)	145 (9.1)	37 (8.9)	19 (8.5)	
Anovulation, *n* (%)	193 (12.1)	49 (11.8)	33 (14.8)	
Male factor, *n* (%)	387 (24.2)	110 (26.5)	49 (22.0)	
Unexplained infertility, *n* (%)	391 (24.4)	88 (21.2)	50 (22.4)	
Protocol of in vitro fertilization				0.332
Natural cycle, *n* (%)	467 (29.2)	115 (27.7)	57 (25.6)	
Hormone replacement therapy cycle, *n* (%)	336 (21.0)	74 (17.8)	48 (21.5)	
Ovulation induction cycle, *n* (%)	76 (4.7)	17 (4.1)	15 (6.7)	
Controlled ovarian hyperstimulation cycle, *n* (%)	722 (45.1)	209 (50.4)	103 (46.2)	
Type of embryo transfer				0.159
Fresh embryo, *n* (%)	722 (45.1)	209 (50.4)	103 (46.2)	
Frozen-thaw embryo, *n* (%)	879 (54.9)	206 (49.6)	120 (53.8)	
Stage of transferred embryo				0.322
Cleavage embryo, *n* (%)	1007 (62.9)	277 (66.7)	145 (65.0)	
Blastocyst, *n* (%)	594 (37.1)	138 (33.3)	78 (35.0)	
Number of embryos transferred, *n*	1.7 ± 0.5	1.7 ± 0.5	1.7 ± 0.5	0.294

* Pairwise comparison with normal BP group significant at *p* < 0.05 level.

**Table 2 jcm-12-00121-t002:** Adverse pregnancy outcomes of the normotension, elevated BP and stage 1 hypertension groups.

	Normal Blood Pressure (*n* = 1601)	Elevated Blood Pressure (*n* = 415)	Stage 1 Hypertension (*n* = 223)	*p*
Hypertensive disorders of pregnancy, *n* (%)	225 (14.1)	58 (14.0)	50 (22.4) *^,∆^	0.004
Preeclampsia, *n* (%)	171 (10.7)	40 (9.6)	36 (16.1) *^,∆^	0.031
Preterm delivery, *n* (%)	320 (20.0)	74 (17.8)	51 (22.9)	0.308
Low birth weight, *n* (%)	290 (18.1)	66 (15.9)	45 (20.2)	0.375
Small for gestational age, *n* (%)	78 (5.5)	19 (5.2)	14 (7.4)	0.45

* Pairwise comparison with normotension group significant at *p* < 0.05 level. ^∆^ Pairwise comparison with elevated BP group significant at *p* < 0.05 level.

**Table 3 jcm-12-00121-t003:** Multiple logistic regression analysis for the association of pre-pregnancy blood pressure with adverse pregnancy outcomes.

	Normal Blood Pressure (*n* = 1601)	Elevated Blood Pressure (*n* = 415)	Stage 1 Hypertension (*n* = 223)
HDP			
Crude OR (95% CI)	1	0.97 (0.71–1.332)	1.77 (1.25–2.50)
Adjusted OR ^a^ (95% CI)	1	0.88 (0.64–1.21)	1.65 (1.16–2.35)
Preeclampsia			
Crude OR (95% CI)	1	0.89 (0.62–1.28)	1.61 (1.09–2.38)
Adjusted OR ^a^ (95% CI)	1	0.83 (0.57–1.20)	1.52 (1.02–2.26)
Preterm birth			
Crude OR (95% CI)	1	0.87 (0.66–1.15)	1.19 (0.85–1.66)
Adjusted OR ^a^ (95% CI)	1	0.87 (0.65–1.19)	1.06 (0.74–1.54)
Low birth weight			
Crude OR (95% CI)	1	0.86 (0.64–1.15)	1.14 (0.81–1.62)
Adjusted OR ^a^ (95% CI)	1	0.89 (0.64–1.23)	1.06 (0.71–1.56)
Small for gestational age			
Crude OR (95% CI)	1	0.94 (0.56–1.57)	1.31 (0.73–2.35)
Adjusted OR ^a^ (95% CI)	1	1.01 (0.60–1.70)	1.26 (0.69–2.30)

BP, blood pressure; HDP, hypertensive disorders of pregnancy; OR, odds ratio; CI, confidence interval; ^a^ Adjusted for maternal age (continuous), pre-pregnancy body mass index (BMI) (continuous), parity, twin pregnancy, previous histories of diabetes mellitus, etiology of infertility, protocol of in vitro fertilization (IVF), type of embryo transfer, stage of transferred embryo, and number of embryos transferred.

**Table 4 jcm-12-00121-t004:** Maternal outcomes of the study population by pre-pregnancy systolic blood pressure.

Systolic Blood Pressure, mmHg	<100 (*n* = 141)	100~109 (*n* = 566)	110~119 (*n* = 928)	120~129 (*n* = 499)	130~139 (*n* = 105)
HDP					
*n* (%)	14 (9.9)	77 (13.6)	138 (14.9)	75 (15.0)	29 (27.6)
Crude OR (95% CI)	1	1.43 (0.78–2.61)	1.59 (0.89–2.83)	1.61 (0.88–2.94)	3.46 (1.72–6.96)
Adjusted OR ^a^ (95% CI)	1	1.18 (0.63–2.19)	1.27 (0.70–2.31)	1.20 (0.64–2.25)	2.24 (1.08–4.64)
Preeclampsia					
*n* (%)	14 (9.9)	57 (10.1)	102 (11.0)	50 (10.0)	24 (22.9)
Crude OR (95% CI)	1	1.02 (0.55–1.88)	1.12 (0.62–2.02)	1.01 (0.54–1.89)	2.69 (1.31–5.50)
Adjusted OR ^a^ (95% CI)	1	1.02 (0.54–1.92)	1.20 (0.66–2.19)	1.08 (0.57–2.04)	2.50 (1.19–5.21)

BP, blood pressure; HDP, hypertensive disorders of pregnancy; OR, odds ratio; CI, confidence interval; ^a^ Adjusted for maternal age (continuous), pre-pregnancy body mass index (BMI) (continuous), parity, twin pregnancy, previous histories of diabetes mellitus, etiology of infertility, protocol of in vitro fertilization (IVF), type of embryo transfer, stage of transferred embryo, and number of embryos transferred.

**Table 5 jcm-12-00121-t005:** Maternal outcomes of the study population by pre-pregnancy diastolic blood pressure.

Diastolic Blood Pressure, mmHg	<70 (*n* = 1046)	70~74 (*n* = 611)	75~79 (*n* = 409)	80~84 (*n* = 123)	85~89 (*n* = 50)
HDP					
*n* (%)	134 (12.8)	95 (15.5)	69 (16.9)	20 (16.3)	15 (30)
Crude OR (95% CI)	1	1.25 (0.94–1.67)	1.38 (1.01–1.89)	1.32 (0.79–2.21)	2.92 (1.55–5.49)
Adjusted OR ^a^ (95% CI)	1	1.18 (0.87–1.59)	1.32 (0.95–1.85)	1.14 (0.67–1.95)	2.58 (1.30–5.16)
Preeclampsia					
*n* (%)	101 (9.7)	74 (12.1)	49 (12.0)	13 (10.6)	10 (20)
Crude OR (95% CI)	1	1.29 (0.94–1.77)	1.27 (0.89–1.83)	1.11 (0.60–2.04)	2.34 (1.14–4.82)
Adjusted OR ^a^ (95% CI)	1	1.26 (0.90–1.75)	1.24 (0.85–1.81)	0.99 (0.53–1.87)	2.10 (1.01–4.56)

BP, blood pressure; HDP, hypertensive disorders of pregnancy; OR, odds ratio; CI, confidence interval; ^a^ Adjusted for maternal age (continuous), pre-pregnancy body mass index (BMI) (continuous), parity, twin pregnancy, previous histories of diabetes mellitus, etiology of infertility, protocol of in vitro fertilization (IVF), type of embryo transfer, stage of transferred embryo, and number of embryos transferred.

## Data Availability

Data available on request from the authors.

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
