# Peer review of "Pregnancy Outcomes in Females with Stage 1 Hypertension and Elevated Blood Pressure Undergoing In Vitro Fertilization and Embryo Transfer"

_jcm, 2022, doi:10.3390/jcm12010121_

Round 1

Reviewer 1 Report

In this retrospective cohort study, the authors (Dr. Chen and colleagues) evaluated the gestationsl outcomes of a large group of 2239 with stage 1 hypertension and elevated blood pressure undergoing in vitro fertilization and embryo transfer. A total of 415 (18.5%) females had elevated blood pressure and 223 (10.0%) had stage 1 hypertension. Moreover, the study shows that females with stage 1 hypertension had increased risks for hypertensive disorders in pregnancy. Authors concluded that stage 1 hypertension before undergoing in vitro fertilization and embryo transfer was an independent risk factor for hypertensive disorders in pregnancy and preeclampsia. Here various minor/major comments
1.    General observation, the work should be revised by a language expert given the presence of several writing errors and typos

2.    AAbstract instead of “415 (18.5%)” I suggest the following form: “18.5% (415/2239)”. Same observation can be done at page 3, lines 129-130, page 5 line 150 etc

3.    I suggest using “females” in substation of “women”

4.    Page 1, line 38 “in vitro” should be in italics, please revise the text

5.    All upperscore references should be moved after the periods/commas.

6.    Page 1 Lines 45-46 additional factors related to adverse pregnancy outcomes comprise, mainly, genetic alterations (PMID: 15045131), lifestyle factors  (doi: 10.11604/pamj.2016.25.111.8739), infections (PMID: 34970247), immunological alterations (DOI: 10.3389/fphar.2022.895254). For completeness of information, these additional factors related to adverse pregnancy outcomes should be mentioned.

7.    I also suggest including a couple of words on the risk of negative pregnancy outcomes following in vitro fertilization, with particular attention on the role of male factor (doi: 10.1016/j.fertnstert.2015.04.006 and doi: 10.1007/s10815-021-02259-1). In this context, males inclusion criteria in terms of spermiogram, and/or additional clinical data should be detailed and statistically evaluated with the authors findings obtained from the females. It cannot be excluded that the male component might have influenced the gestation outcomes of the females enrolled in the study. Notoriously, males belonging to couples undergoing IVF-ET play an important role in negatively impacting the pregnancy outcomes, with serious consequences such as embryo loss and or preeclampsia (https://www.frontiersin.org/articles/10.3389/fphys.2018.01870/full). If no male data are available, this should be stated among study limitations

8.    Please include references supporting the applied statistical tests

9.    Page 3, line 131 “Table 1 summarizes”. In table, I suggest avoiding acronyms

10.    Please explain the meaning of “Small for gestational “ in table 2

11.    Fig 2 since the reported data are not longitudinal, I suggest replacing line graph for bar graph.

12.    Page 8 line 227 “meta-analyses”

13.    Page 8, lines 252-253 is this assumption being supported by the clinical data obtained from the females? If yes, it should be underlined

14.    Page 8, lines 256 “differently from”

15.    Page 8, lines 256 Given the “previous studies” being mentioned, supporting references should be included at the end of the sentence

Author Response

Dear Dr:

   Thank you for your suggestions. We carefully considered your suggestions and made these revisions.

  1. General observation, the work should be revised by a language expert given the presence of several writing errors and typos 

Reply: The manuscript was revised by a language expert.

2.    Abstract instead of “415 (18.5%)” I suggest the following form: “18.5% (415/2239)”. Same observation can be done at page 3, lines 129-130, page 5 line 150 etc Reply: The expressions were changed according the suggestion of the reviewer.

3.    I suggest using “females” in substation of “women”  Reply: we replaced “women” by “females”.

4.    Page 1, line 38 “in vitro” should be in italics, please revise the text
     Reply: “in vitro” were written in italics
5.    All upperscore references should be moved after the periods/commas.
     Reply: All upperscore references were moved after the periods/commas.
6.    Page 1 Lines 45-46 additional factors related to adverse pregnancy outcomes comprise, mainly, genetic alterations (PMID: 15045131), lifestyle factors  (doi: 10.11604/pamj.2016.25.111.8739), infections (PMID: 34970247), immunological alterations (DOI: 10.3389/fphar.2022.895254). For completeness of information, these additional factors related to adverse pregnancy outcomes should be mentioned. Reply: In this study, we focused on the effect of pre-pregnancy blood pressure on the outcomes. In the multivariate analysis, we included the confounding factors (body mass index, age ….) which have been confirmed to be risk factors of adverse outcomes. There were only a few studies about lifestyle factors, infections and immunological alterations. So we did not include these factors.

7.    I also suggest including a couple of words on the risk of negative pregnancy outcomes following in vitro fertilization, with particular attention on the role of male factor (doi: 10.1016/j.fertnstert.2015.04.006 and doi: 10.1007/s10815-021-02259-1). In this context, males inclusion criteria in terms of spermiogram, and/or additional clinical data should be detailed and statistically evaluated with the authors findings obtained from the females. It cannot be excluded that the male component might have influenced the gestation outcomes of the females enrolled in the study. Notoriously, males belonging to couples undergoing IVF-ET play an important role in negatively impacting the pregnancy outcomes, with serious consequences such as embryo loss and or preeclampsia (https://www.frontiersin.org/articles/10.3389/fphys.2018.01870/full). If no male data are available, this should be stated among study limitations 
    Reply: Male factor infertility has been shown to be a risk factor for adverse outcomes. In table 1, the etiology of infertility included male factor. We added the statement in the discussion “ studies have also suggested that endometriosis, male factor infertility, hormone replacement therapy cycles and frozen embryo transfer are associated with increased risk of adverse pregnancy outcomes. In our study, stage 1 hypertension was independently associated with the risk of HDP and preeclampsia after adjustment for these confounding factors.”
8.    Please include references supporting the applied statistical tests Reply: We included the Ref 15 to support the applied statistical tests.

9.    Page 3, line 131 “Table 1 summarizes”. In table, I suggest avoiding acronyms Reply: the acronyms in tables were replaced by the whole word.

10.    Please explain the meaning of “Small for gestational “ in table 2Reply: “Small for gestational age” was explained in the part of  “Outcomes”(page 2)

11.    Fig 2 since the reported data are not longitudinal, I suggest replacing line graph for bar graph.Reply: The line graph in Fig 2 was replaced by bar graph.
12.    Page 8 line 227 “meta-analyses”Reply: “meta-analysis” was replaced by “meta-analyses”.

13.    Page 8, lines 252-253 is this assumption being supported by the clinical data obtained from the females? If yes, it should be underlinedReply: We added Ref 8 to support “ blood pressure declines during early pregnancy”, but “ females with elevated BP during early pregnancy might have higher BP before pregnancy” was our assumption.

14.    Page 8, lines 256 “differently from”Reply: “different from this study”  was changed to “by contrast”.

15.    Page 8, lines 256 Given the “previous studies” being mentioned, supporting references should be included at the end of the sentenceReply: We added Ref 14,15,18,21 to support the sentence.

Reviewer 2 Report

The strong side of the paper is large study group and quite smartly conducted study. Unfortunately, it is weaken due to some methodological mistakes: authors mention about patient's BMI but I found not a single word on correlation between BMI and pressure values. Authors also don't mention on circulation diseases in the studied group. Those drawbacks should be improved.

Author Response

Dear Dr:

   Thank you for your suggestions. We carefully considered your suggestions and made these revisions.

‘authors mention about patient's BMI but I found not a single word on correlation between BMI and pressure values’.

Reply:Females with elevated BP and stage 1 hypertension had significantly higher pre-pregnancy BMI than women with normal BP.  We analyzed the differences among the three BP levels, but we did not analyze the linear relationship between BP and BMI. There was linear relationship between BP and BMI, but we did show the results since the purpose of this study was to determine the relationship between BP and adverse outcomes. BMI is a risk factor for adverse outcomes, so we included BMI in multivariate analysis.

Authors also don't mention on circulation diseases in the studied group. Those drawbacks should be improved.

Reply: We add the statement in ‘ Limitations’. ‘We did not collect data on cardiovascular and renal diseases that can cause adverse pregnancy outcomes because these diseases are rare in females undergoing IVF-ET.’

Round 2

Reviewer 2 Report

Authors answers on my previous questions in clear and satisfactory way